# Joint Optimization of Tree-based Index and Deep Model for Recommender Systems

**Han Zhu[1], Daqing Chang[1], Ziru Xu[1,2], Pengye Zhang[1]**
[1]Alibaba Group
[2]School of Software, Tsinghua University
Beijing, China
{zhuhan.zh, daqing.cdq, ziru.xzr, pengye.zpy}@alibaba-inc.com

**Xiang Li, Jie He, Han Li, Jian Xu, Kun Gai**
Alibaba Group
Beijing, China
{yushi.lx, jay.hj, lihan.lh, xiyu.xj, jingshi.gk}@alibaba-inc.com

## Abstract

Large-scale industrial recommender systems are usually confronted with computational problems due to the enormous corpus size. To retrieve and recommend the most relevant items to users under response time limits, resorting to an efficient index structure is an effective and practical solution. The previous work Tree-based Deep Model (TDM) [34] greatly improves recommendation accuracy using tree index. By indexing items in a tree hierarchy and training a user-node preference prediction model satisfying a max-heap like property in the tree, TDM provides logarithmic computational complexity w.r.t. the corpus size, enabling the use of arbitrary advanced models in candidate retrieval and recommendation.

In tree-based recommendation methods, the quality of both the tree index and the user-node preference prediction model determines the recommendation accuracy for the most part. We argue that the learning of tree index and preference model has interdependence. Our purpose, in this paper, is to develop a method to jointly learn the index structure and user preference prediction model. In our proposed joint optimization framework, the learning of index and user preference prediction model are carried out under a unified performance measure. Besides, we come up with a novel hierarchical user preference representation utilizing the tree index hierarchy. Experimental evaluations with two large-scale real-world datasets show that the proposed method improves recommendation accuracy significantly. Online A/B test results at a display advertising platform also demonstrate the effectiveness of the proposed method in production environments.

## 1   Introduction

Recommendation problem is basically to retrieve a set of most relevant or preferred items for each user request from the entire corpus. In the practice of large-scale recommendation, the algorithm design should strike a balance between accuracy and efficiency. In corpus with tens or hundreds of millions of items, methods that need to linearly scan each item's preference score for each single user request are not computationally tractable. To solve the problem, index structure is commonly used to accelerate the retrieval process. In early recommender systems, item-based collaborative

filtering (Item-CF) along with the inverted index is a popular solution to overcome the calculation barrier [18]. However, the scope of candidate set is limited, because only those items similar to user's historical behaviors can be ultimately recommended.

In recent days, vector representation learning methods [27, 16, 26, 5, 22, 2] have been actively researched. This kind of methods can learn user and item's vector representations, the inner-product of which represents user-item preference. For systems that use vector representation based methods, the recommendation set generation is equivalent to the k-nearest neighbor (kNN) search problem. Quantization-based index [19, 14] for approximate kNN search is widely adopted to accelerate the retrieval process. However, in the above solution, the vector representation learning and the kNN search index construction are optimized towards different objectives individually. The objective divergence leads to suboptimal vector representations and index structure [4]. An even more important problem is that the dependence on vector kNN search index requires an inner-product form of user preference modeling, which limits the model capability [10]. Models like Deep Interest Network [32], Deep Interest Evolution Network [31] and xDeepFM [17], which have been proven to be effective in user preference prediction, could not be used to generate candidates in recommendation.

In order to break the inner-product form limitation and make arbitrary advanced user preference models computationally tractable to retrieve candidates from the entire corpus, the previous work Tree-based Deep Model (TDM) [34] creatively uses tree structure as index and greatly improves the recommendation accuracy. TDM uses a tree index to organize items, and each leaf node in the tree corresponds to an item. Like a max-heap, TDM assumes that each user-node preference equals to the largest one among the user's preference over all children of this node. In the training stage, a user-node preference prediction model is trained to fit the max-heap like preference distribution. Unlike vector kNN search based methods where the index structure requires an inner-product form of user preference modeling, there is no restriction on the form of preference model in TDM. And in prediction, preference scores given by the trained model are used to perform layer-wise beam search in the tree index to retrieve the candidate items. The time complexity of beam search in tree index is logarithmic w.r.t. the corpus size and no restriction is imposed on the model structure, which is a prerequisite to make advanced user preference models feasible to retrieve candidates in recommendation.

The index structure plays different roles in kNN search based methods and tree-based methods. In kNN search based methods, the user and item's vector representations are learnt first, and the vector search index is built then. While in tree-based methods, the tree index's hierarchy also affects the retrieval model training. Therefore, how to learn the tree index and user preference model jointly is an important problem. Tree-based method is also an active research topic in literature of extreme classification [29, 1, 24, 11, 8, 25], which is sometimes considered the same as recommendation [12, 25]. In the existing tree-based methods, the tree structure is learnt for a better hierarchy in the sample or label space. However, the objective of sample or label partitioning task in the tree learning stage is not fully consistent with the ultimate target, i.e., accurate recommendation. The inconsistency between objectives of index learning and prediction model training leads the overall system to a suboptimal status. To address this challenge and facilitate better cooperation of tree index and user preference prediction model, we focus on developing a way to simultaneously learn the tree index and user preference prediction model by optimizing a unified performance measure.

The main contributions of this paper are: 1) We propose a joint optimization framework to learn the tree index and user preference prediction model in tree-based recommendation, where a unified performance measure, i.e., the accuracy of user preference prediction is optimized; 2) We demonstrate that the proposed tree learning algorithm is equivalent to the weighted maximum matching problem of bipartite graph, and give an approximate algorithm to learn the tree; 3) We propose a novel method that makes better use of tree index to generate hierarchical user representation, which can help learn more accurate user preference prediction model; 4) We show that both the tree index learning and hierarchical user representation can improve recommendation accuracy, and these two modules can even mutually improve each other to achieve more significant performance promotion.

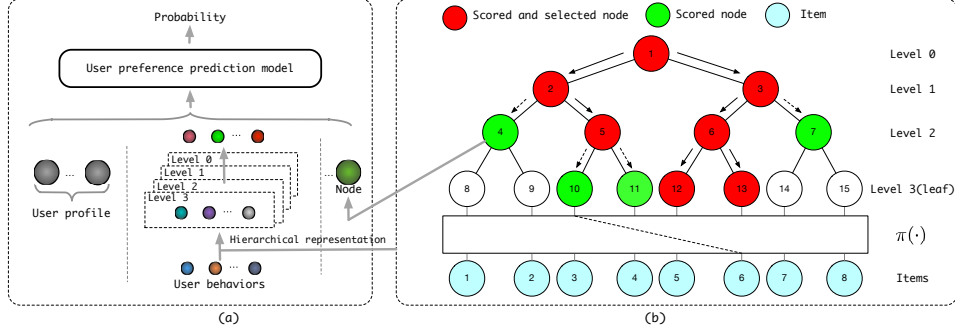

Figure 1: Tree-based deep recommendation model. (a) User preference prediction model. We firstly hierarchically abstract the user behaviors with nodes in corresponding levels. Then the abstract user behaviors and the target node together with the other feature such as the user profile are used as the input of the model. (b) Tree hierarchy. Each item is firstly assigned to a different leaf node with a projection function $\pi(\cdot)$. In retrieval stage, items that assigned to the red nodes in the leaf level are selected as the candidate set.

## 2 Joint Optimization of Tree-based Index and Deep Model

In this section, we firstly give a brief review of TDM [34] to make this paper self-contained. Then we propose the joint learning framework of the tree-based index and deep model. In the last subsection, we specify the hierarchical user preference representation used in model training.

### 2.1 Tree-based Deep Recommendation Model

In recommender systems with large-scale corpus, how to retrieve candidates effectively and efficiently is a challenging problem. TDM uses a tree as index and proposes a max-heap like probability formulation in the tree, where the user preference for each non-leaf node $n$ in level $l$ is derived as:

$$p^{(l)}(n|u) = \frac{\max_{n_c \in \{n's \text{ children in level } l+1\}} p^{(l+1)}(n_c|u)}{\alpha^{(l)}} \tag{1}$$

where $p^{(l)}(n|u)$ is the ground truth probability that the user $u$ prefers the node $n$. $\alpha^{(l)}$ is a level normalization term. The above formulation means that the ground truth user-node probability on a node equals to the maximum user-node probability of its children divided by a normalization term. Therefore, the top-k nodes in level $l$ must be contained in the children of top-k nodes in level $l-1$, and the retrieval for top-k leaf items can be restricted to recursive top-k nodes retrieval top-down in each level without losing the accuracy. Based on this, TDM turns the recommendation task into a hierarchical retrieval problem, where the candidate items are selected gradually from coarse to fine. The candidate generating process of TDM is shown in Fig 1.

Each item is firstly assigned to a leaf node in the tree hierarchy $\mathcal{T}$. A layer-wise beam search strategy is carried out as shown in Fig1(b). For level $l$, only the children of nodes with top-k probabilities in level $l-1$ are scored and sorted to pick $k$ candidate nodes in level $l$. This process continues until $k$ leaf items are reached. User features combined with the candidate node are used as the input of the prediction model $\mathcal{M}$ (e.g. fully-connected networks) to get the preference probability, as shown in Fig 1(a). With tree index, the overall retrieval complexity for a user request is reduced from linear to logarithmic w.r.t. the corpus size, and there is no restriction on the preference model structure. This makes TDM break the inner-product form of user preference modeling restriction brought by vector kNN search index and enable arbitrary advanced deep models to retrieve candidates from the entire corpus, which greatly raises the recommendation accuracy.

### 2.2 Joint Optimization Framework

Derive the training set that has $n$ samples as $\{(u^{(i)}, c^{(i)})\}_{i=1}^{n}$, in which the $i$-th pair $(u^{(i)}, c^{(i)})$ means the user $u^{(i)}$ is interested in the target item $c^{(i)}$. For $(u^{(i)}, c^{(i)})$, tree hierarchy $\mathcal{T}$ determines the path that prediction model $\mathcal{M}$ should select to achieve $c^{(i)}$ for $u^{(i)}$. We propose to jointly learn $\mathcal{M}$ and

**Algorithm 1:** Joint learning framework of the tree index and deep model

---

**Input:** Loss function $\mathcal{L}(\theta, \pi)$, initial deep model $\mathcal{M}$ and initial tree $\mathcal{T}$
 1: **for** $t = 0, 1, 2 \dots$ **do**
 2:    Solve $\min_\theta \mathcal{L}(\theta, \pi)$ by optimizing the model $\mathcal{M}$.
 3:    Solve $\max_\pi -\mathcal{L}(\theta, \pi)$ by optimizing the tree hierarchy with Algorithm 2
 4: **end for**
**Output:** Learned model $\mathcal{M}$ and tree $\mathcal{T}$

---

$\mathcal{T}$ under a global loss function. As we will see in experiments, jointly optimizing $\mathcal{M}$ and $\mathcal{T}$ could improve the ultimate recommendation accuracy.

Given a user-item pair $(u, c)$, denote $p\left(\pi(c)|u; \pi\right)$ as user $u$'s preference probability over leaf node $\pi(c)$ where $\pi(\cdot)$ is a projection function that projects an item to a leaf node in $\mathcal{T}$. Note that $\pi(\cdot)$ completely determines the tree hierarchy $\mathcal{T}$, as shown in Fig 1(b). And optimizing $\mathcal{T}$ is actually optimizing $\pi(\cdot)$. The model $\mathcal{M}$ estimates the user-node preference $\hat{p}\left(\pi(c)|u; \theta, \pi\right)$, given $\theta$ as model parameters. If the pair $(u, c)$ is a positive sample, we have the ground truth preference $p\left(\pi(c)|u; \pi\right) = 1$ following the multi-class setting [5, 2]. According to the max-heap property, the user preference probability of all $\pi(c)$'s ancestor nodes, i.e., $\{p(b_j(\pi(c))|u; \pi)\}_{j=0}^{l_{max}}$ should also be 1, in which $b_j(\cdot)$ is the projection from a node to its ancestor node in level $j$ and $l_{max}$ is the max level in $\mathcal{T}$. To fit such a user-node preference distribution, the global loss function is formulated as

$$\mathcal{L}(\theta, \pi) = -\sum_{i=1}^{n} \sum_{j=0}^{l_{max}} \log \hat{p}\left(b_j(\pi(c^{(i)}))|u^{(i)}; \theta, \pi\right), \tag{2}$$

where we sum up the negative logarithm of predicted user-node preference probability on all the positive training samples and their ancestor user-node pairs as the global empirical loss.

Optimizing $\pi(\cdot)$ is a combinational optimization problem, which can hardly be simultaneously optimized with $\theta$ using gradient-based algorithms. To conquer this, we propose a joint learning framework as shown in Algorithm 1. It alternatively optimizes the loss function (2) with respect to the user preference model and the tree hierarchy. The consistency of the training loss in model training and tree learning promotes the convergence of the framework. Actually, Algorithm 1 surely converges if both the model training and tree learning decrease the value of (2) since $\{\mathcal{L}(\theta_t, \pi_t)\}$ is a decreasing sequence and lower bounded by 0. In model training, $\min_\theta \mathcal{L}(\theta, \pi)$ is to learn a user-node preference model for all levels, which can be solved by popular optimization algorithms for neural networks such as SGD[3], Adam[15]. In the normalized user preference setting [5, 2], since the number of nodes increases exponentially with the node level, Noise-contrastive estimation[7] is an alternative to estimate $\hat{p}\left(b_j(\pi(c))|u; \theta, \pi\right)$ to avoid calculating the normalization term by sampling strategy. The task of tree learning is to solve $\max_\pi -\mathcal{L}(\theta, \pi)$ given $\theta$. $\max_\pi -\mathcal{L}(\theta, \pi)$ equals to the maximum weighted matching problem of bipartite graph that consists of items in the corpus $\mathcal{C}$ and the leaf nodes of $\mathcal{T}$[2]. The detailed proof is shown in the supplementary material.

Traditional algorithms for assignment problem such as the classic Hungarian algorithm are hard to apply for large corpus because of their high complexities. Even for the naive greedy algorithm that greedily chooses the unassigned edge with the largest weight, a big weight matrix needs to be computed and stored in advance, which is not acceptable. To conquer this issue, we propose a segmented tree learning algorithm.

Instead of assigning items directly to leaf nodes, we achieve this step-by-step from the root node to the leaf level. Given a projection $\pi$ and the $k$-th item $c_k$ in the corpus, denote

$$\mathcal{L}_{c_k}^{s,e}(\pi) = \sum_{(u,c) \in \mathcal{A}_k} \sum_{j=s}^{e} \log \hat{p}\left(b_j(\pi(c))|u; \theta, \pi\right),$$

where $\mathcal{A}_k = \{(u^{(i)}, c^{(i)})|c^{(i)} = c_k\}_{i=1}^{n}$ is the set of training samples whose target item is $c_k$, $s$ and $e$ are the start and end level respectively. We firstly maximize $\sum_{k=1}^{|\mathcal{C}|} \mathcal{L}_{c_k}^{1,d}(\pi)$ w.r.t. $\pi$, which is

**Algorithm 2:** Tree learning algorithm
---
**Input:** Gap $d$, max tree level $l_{\max}$, original projection $\pi_{old}$
**Output:** Optimized projection $\pi_{new}$
 1: Set current level $l \leftarrow d$, initialize $\pi_{new} \leftarrow \pi_{old}$
 2: **while** $d > 0$ **do**
 3:    **for** each node $n_i$ in level $l - d$ **do**
 4:       Denote $\mathcal{C}_{n_i}$ as the item set that $\forall c \in \mathcal{C}_{n_i}, b_{l-d}(\pi_{new}(c)) = n_i$
 5:       Find $\pi^*$ that maximize $\sum_{c \in \mathcal{C}_{n_i}} \mathcal{L}_c^{l-d+1,l}(\pi)$, s.t. $\forall c \in \mathcal{C}_{n_i}, b_{l-d}(\pi^*(c)) = n_i$
 6:       Update $\pi_{new}$. $\forall c \in \mathcal{C}_{n_i}, \pi_{new}(c) \leftarrow \pi^*(c)$
 7:    **end for**
 8:    $d \leftarrow \min(d, l_{max} - l)$
 9:    $l \leftarrow l + d$
10: **end while**
---

equivalent to assign all the items to nodes in level $d$. For a complete binary tree $\mathcal{T}$ with max level $l_{max}$, each node in level $d$ is assigned with no more than $2^{l_{max}-d}$ items. This is also a maximum matching problem which can be efficiently solved by a greedy algorithm, since the number of possible locations for each item is largely decreased if $d$ is well chosen (e.g. for $d = 7$, the number is $2^d = 128$). Denote the found optimal projection in this step as $\pi^*$. Then, we successively maximize $\sum_{k=1}^{|\mathcal{C}|} \mathcal{L}_{c_k}^{d+1,2d}(\pi)$ under the constraint that $\forall c \in \mathcal{C}, b_d(\pi(c)) = b_d(\pi^*(c))$, which means keeping each item's corresponding ancestor node in level $d$ unchanged. The recursion stops until each item is assigned to a leaf node. The proposed algorithm is detailed in Algorithm 2.

In line 5 of Algorithm 2, we use a greedy algorithm with rebalance strategy to solve the sub-problem. Each item $c \in \mathcal{C}_{n_i}$ is firstly assigned to the child of $n_i$ in level $l$ with largest weight $\mathcal{L}_c^{l-d+1,l}(\cdot)$. Then, a rebalance process is applied to ensure that each child is assigned with no more than $2^{l_{max}-l}$ items. The detailed implementation of Algorithm 2 is given in the supplementary material.

## 2.3 Hierarchical User Preference Representation

As shown in Section 2.1, TDM is a hierarchical retrieval model to generate the candidate items hierarchically from coarse to fine. In retrieval, a layer-wise top-down beam search is carried out through the tree index by the user preference prediction model $\mathcal{M}$. Therefore, $\mathcal{M}'s$ task in each level are heterogeneous. Based on this, a level-specific input of $\mathcal{M}$ is necessary to raise the recommendation accuracy.

A series of related work [30, 6, 18, 16, 32, 33, 34] has shown that the user's historical behaviors play a key role in predicting the user's interests. However, in our tree-based approach we could even enlarge this key role in a novel and effective way. Given a user behavior sequence $\boldsymbol{c} = \{c_1, c_2, \cdots, c_m\}$ where $c_i$ is the $i$-th item the user interacts, we propose to use $\boldsymbol{c}^l = \{b_l(\pi(c_1)), b_l(\pi(c_2)), \cdots, b_l(\pi(c_m))\}$ as user's behavior feature in level $l$. $\boldsymbol{c}^l$ together with the target node and other possible features such as user profile are used as the input of $\mathcal{M}$ in level $l$ to predict the user-node preference, as shown in Fig 1(a). In addition, since each node or item is a one-hot ID feature, we follow the common way to embed them into continuous feature space. In this way, the ancestor nodes of items the user interacts are used as the hierarchical user preference representation. Generally, the hierarchical representation brings two main benefits:

1. **Level independence**. As in the common way, sharing item embeddings between different levels will bring noises in training the user preference prediction model $\mathcal{M}$, because the targets differ for different levels. An explicit solution is to attach an item with an independent embedding for each level. However, this will greatly increase the number of parameters and make the system hard to optimize and apply. The proposed hierarchical representation uses node embeddings in the corresponding level as the input of $\mathcal{M}$, which achieves level independence in training without increasing the number of parameters.

2. **Precise description**. $\mathcal{M}$ generates the candidate items hierarchically through the tree. With the increase of retrieval level, the candidate nodes in each level describe the ultimate recommended items from coarse to fine until the leaf level is reached. The proposed hierarchical user preference representation grasps the nature of the retrieval process and gives a precise description of user

behaviors with nodes in corresponding level, which promotes the predictability of user preference by reducing the confusion brought by too detailed or coarse description. For example, $\mathcal{M}$'s task in upper levels is to coarsely select a candidate set and the user behaviors are also coarsely described with homogeneous node embeddings in the same upper levels in training and prediction.

Experimental study in both Section 3 and the supplementary material will show the significant effectiveness of the proposed hierarchical representation.

# 3    Experimental Study

We study the performance of the proposed method both offline and online in this section. We firstly compare the overall performance of the proposed method with other baselines. Then we conduct experiments to verify the contribution of each part and convergence of the framework. At last, we show the performance of the proposed method in an online display advertising platform with real traffic.

## 3.1    Experiment Setup

The offline experiments are conducted with two large-scale real-world datasets: 1) **Amazon Books**[3][20, 9], a user-book review dataset made up of product reviews from Amazon. Here we use its largest subset Books; 2) **UserBehavior**[4][34], a subset of Taobao user behavior data. These two datasets both contain millions of items and the data is organized in user-item interaction form: each user-item interaction consists of user ID, item ID, category ID and timestamp. For the above two datasets, only users with no less than 10 interactions are kept.

To evaluate the performance of the proposed framework, we compare the following methods:

- **Item-CF** [28] is a basic collaborative filtering method and is widely used for personalized recommendation especially for large-scale corpus [18].
- **YouTube product-DNN** [5] is a practical method used in YouTube video recommendation. It's the representative work of vector kNN search based methods. The inner-product of the learnt user and item's vector representation reflects the preference. And we use the exact kNN search to retrieve candidates in prediction.
- **HSM** [21] is the hierarchical softmax model. It adopts the multiplication of layer-wise conditional probabilities to get the normalized item preference probability.
- **TDM** [34] is the tree-based deep model for recommendation. It enables arbitrary advanced models to retrieve user interests using the tree index. We use the proposed basic DNN version of TDM without tree learning and attention.
- **DNN** is a variant of TDM without tree index. The only difference is that it directly learns a user-item preference model and linearly scan all items to retrieve the top-k candidates in prediction. It's computationally intractable in online system but a strong baseline in offline comparison.
- **JTM** is the proposed joint learning framework of the tree index and user preference prediction model. **JTM-J** and **JTM-H** are two variants. **JTM-J** jointly optimizes the tree index and user preference prediction model without the proposed hierarchical representation in Section 2.3. And **JTM-H** adopts the hierarchical representation but use the fixed initial tree index without tree learning.

Following TDM [34], we split users into training, validation and testing sets disjointly. Each user-item interaction in training set is a training sample, and the user's behaviors before the interaction are the corresponding features. For each user in validation and testing set, we take the first half of behaviors along the time line as known features and the latter half as ground truth.

Taking advantage of TDM's open source work[5], we implement all methods in Alibaba's deep learning platform X-DeepLearning (XDL). HSM, DNN and JTM adopt the same user preference prediction model with TDM. We deploy negative sampling for all methods except Item-CF and use the same negative sampling ratio. 100 negative items in Amazon Books and 200 in UserBehavior are

sampled for each training sample. HSM, TDM and JTM require an initial tree in advance of training process. Following TDM, we use category information to initialize the tree structure where items from the same category aggregate in the leaf level. More details and codes about data pre-processing and training are listed in the supplementary material.

Precision, Recall and F-Measure are three general metrics and we use them to evaluate the performance of different methods. For a user $u$, suppose $\mathcal{P}_u$ ($|\mathcal{P}_u| = M$) is the recalled set and $\mathcal{G}_u$ is the ground truth set. The equations of three metrics are

$$\text{Precision@}M(u) = \frac{|\mathcal{P}_u \cap \mathcal{G}_u|}{|\mathcal{P}_u|}, \ \text{Recall@}M(u) = \frac{|\mathcal{P}_u \cap \mathcal{G}_u|}{|\mathcal{G}_u|}$$

$$\text{F-Measure@}M(u) = \frac{2 * \text{Precision@}M(u) * \text{Recall@}M(u)}{\text{Precision@}M(u) + \text{Recall@}M(u)}$$

The results of each metric are averaged across all users in the testing set, and the listed values are the average of five different runs.

### 3.2 Comparison Results

Table 1 exhibits the results of all methods in two datasets. It clearly shows that our proposed JTM outperforms other baselines in all metrics. Compared with the previous best model DNN in two datasets, JTM achieves $45.3\%$ and $9.4\%$ recall lift in Amazon Books and UserBehavior respectively.

Table 1: Comparison results of different methods in Amazon Books and UserBehavior ($M = 200$).

| Method | Amazon Books | | | UserBehavior | | |
|---|---|---|---|---|---|---|
| | Precision | Recall | F-Measure | Precision | Recall | F-Measure |
| Item-CF | 0.52% | 8.18% | 0.92% | 1.56% | 6.75% | 2.30% |
| YouTube product-DNN | 0.53% | 8.26% | 0.93% | 2.25% | 10.15% | 3.36% |
| HSM | 0.42% | 6.22% | 0.72% | 1.80% | 8.62% | 2.71% |
| TDM | 0.50% | 7.49% | 0.88% | 2.23% | 10.84% | 3.40% |
| DNN | 0.56% | 8.57% | 0.98% | 2.81% | 13.45% | 4.23% |
| JTM-J | 0.51% | 7.60% | 0.89% | 2.48% | 11.72% | 3.73% |
| JTM-H | 0.68% | 10.45% | 1.19% | 2.66% | 12.93% | 4.02% |
| JTM | **0.79%** | **12.45%** | **1.38%** | **3.11%** | **14.71%** | **4.68%** |

As mentioned before, though computationally intractable in online system, DNN is a significantly strong baseline for offline comparison. Comparison results of DNN and other methods give insights in many aspects.

Firstly, gap between YouTube product-DNN and DNN shows the limitation of inner-product form. The only difference between these two methods is that YouTube product-DNN uses the inner-product of user and item's vectors to calculate the preference score, while DNN uses a fully-connected network. Such a change brings apparent improvement, which verifies the effectiveness of advanced neural network over inner-product form.

Next, TDM performs worse than DNN with an ordinary but not optimized tree hierarchy. Tree hierarchy takes effect in both training and prediction process. User-node samples are generated along the tree to fit max-heap like preference distribution, and layer-wise beam search is deployed in the tree index when prediction. Without a well-defined tree hierarchy, user preference prediction model may converge to a suboptimal version with confused generated samples, and it's possible to lose targets in the non-leaf levels so that inaccurate candidate sets may be returned. Especially in sparse dataset like Amazon Books, learnt embedding of each node in tree hierarchy is not distinguishable enough so that TDM doesn't perform well than other baselines. This phenomenon illustrates the influence of tree and necessity of tree learning. Additionally, HSM gets much worse results than TDM. This result is consistent with that reported in TDM[34]. When dealing with large corpus, as a result of layer-wise probability multiplication and beam search, HSM cannot guarantee the final recalled set to be optimal.

**By joint learning of tree index and user preference model, JTM outperforms DNN on all metrics in two datasets with much lower retrieval complexity**. More precise user preference predic-

tion model and better tree hierarchy are obtained in JTM, which leads a better item set selection. Hierarchical user preference representation alleviates the data sparsity problem in upper levels, because the feature space of user behavior feature is much smaller while having the same number of samples. And it helps model training in a layer-wise way to reduce the propagation of noises between levels. Besides, tree hierarchy learning makes similar items aggregate in the leaf level, so that the internal level models can get training samples with more consistent and unambiguous distribution. Benefited from the above two reasons, JTM provides better results than DNN.

Results in Table 1 under the dash line indicate the contribution of each part and their joint performance in JTM. Take the recall metric as an example. Compared to TDM in UserBehavior, tree learning and hierarchical representation of user preference brings $0.88\%$ and $2.09\%$ absolute gain separately. Furthermore, $3.87\%$ absolute recall promotion is achieved by the corporation of both optimizations under a unified objective. Similar gain is observed in Amazon Books. The above results clearly show the effectiveness of hierarchical representation and tree learning, as well as the joint learning framework.

**Convergence of Iterative Joint Learning**  Tree hierarchy determines sample generation and search path. A suitable tree would benefit model training and inference a great deal. Fig 2 gives the comparison of clustering-based tree learning algorithm proposed in TDM [34] and our proposed joint learning approach. For fairness, two methods both adopt hierarchical user representation.

Since the proposed tree learning algorithm has the same objective with the user preference prediction model, it has two merits from the results: 1) It can converge to an optimal tree stably; 2) The final recommendation accuracy is higher than the clustering-based method. From Fig 2, we can see that results increase iteratively on all three metrics. Besides, the model stably converges in both datasets, while clustering-based approach ultimately overfits. The above results demonstrate the effectiveness and convergence of iterative joint learning empirically. Some careful readers might have noticed that the clustering algorithm outperforms JTM in the first few iterations. The reason is that the tree learning algorithm in JTM involves a *lazy strategy*, i.e., try to reduce the degree of tree structure change in each iteration (details are given in the supplementary material).

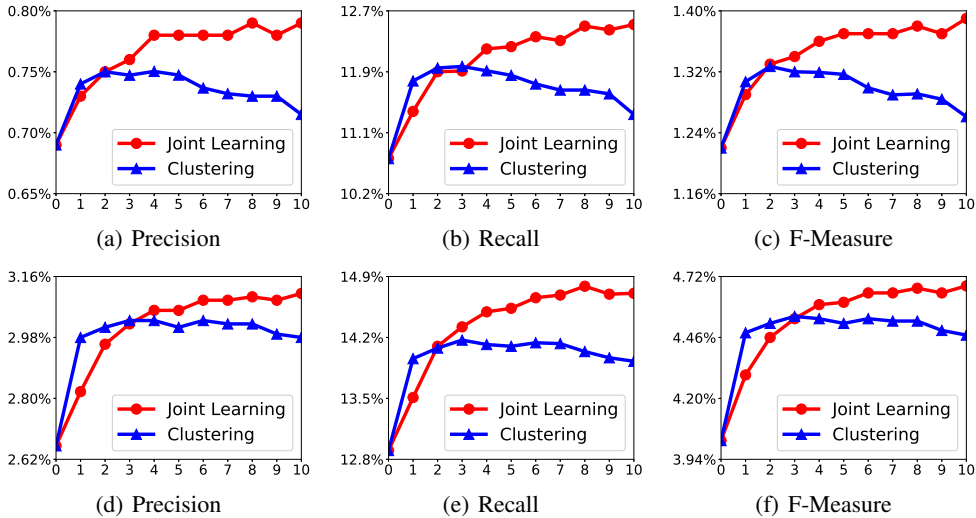

Figure 2: Results of iterative joint learning in two datasets ($M = 200$). 2(a), 2(b), 2(c) are results in Amazon Books and 2(d), 2(e), 2(f) shows the performance in UserBehavior. The horizontal axis of each figure represents the number of iterations.

### 3.3  Online Results

We also evaluate the proposed JTM in production environments: the display advertising scenario of *Guess What You Like* column of Taobao App Homepage. We use click-through rate (CTR) and revenue per mille (RPM) to measure the performance, which are the key performance indicators.

The definitions are:

$$\text{CTR} = \frac{\text{\# of clicks}}{\text{\# of impressions}}, \ \text{RPM} = \frac{\text{Ad revenue}}{\text{\# of impressions}} * 1000$$

In the platform, advertisers bid on plenty of granularities like ad clusters, items, shops, etc. Several simultaneously running recommendation approaches in all granularities produce candidate sets and the combination of them are passed to subsequent stages, like CTR prediction [32, 31, 23], ranking [33, 13], etc. The comparison baseline is such a combination of all running recommendation methods. To assess the effectiveness of JTM, we deploy JTM to replace Item-CF, which is one of the major candidate-generation approaches in granularity of items in the platform. TDM is evaluated in the same way as JTM. The corpus to deal with contains tens of millions of items. Each comparison bucket has 2% of the online traffic, which is big enough considering the overall page view request amount. Table 2 lists the promotion of the two main online metrics. 11.3% growth on CTR exhibits that more precise items have been recommended with JTM. As for RPM, it has a 12.9% improvement, indicating JTM can bring more income for the platform.

Table 2: Online results from Jan 21 to Jan 27, 2019.

| Metric | Baseline | TDM | JTM |
|--------|----------|-------|--------|
| CTR | 0.0% | +5.4% | +11.3% |
| RPM | 0.0% | +7.6% | +12.9% |

## 4  Conclusion

Recommender system plays a key role in various kinds of applications such as video streaming and e-commerce. In this paper, we address an important problem in large-scale recommendation, i.e., how to optimize user representation, user preference prediction and the index structure under a global objective. To the best of our knowledge, JTM is the first work that proposes a unified framework to integrate the optimization of these three key factors. A joint learning approach of the tree index and user preference prediction model is introduced in this framework. The tree index and deep model are alternatively optimized under a global loss function with a novel hierarchical user representation based on the tree index. Both online and offline experimental results show the advantages of the proposed framework over other large-scale recommendation models.

## Acknowledgements

We deeply appreciate Jingwei Zhuo, Mingsheng Long, Jin Li for their helpful suggestions and discussions. Thank Huimin Yi, Yang Zheng and Xianteng Wu for implementing the key components of the training and inference platform. Thank Yin Yang, Liming Duan, Yao Xu, Guan Wang and Yue Gao for necessary supports about online serving.

## Footnotes

*The work is done when she was a student intern in Alibaba Group

[2]For convenience, we assume $\mathcal{T}$ is a given complete binary tree. It is worth mentioning that the proposed algorithm can be naturally extended to multi-way trees.

[3]http://jmcauley.ucsd.edu/data/amazon

[4]http://tianchi.aliyun.com/dataset/dataDetail?dataId=649&userId=1

[5]http://github.com/alibaba/x-deeplearning/tree/master/xdl-algorithm-solution/TDM

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
