[Supplementary Material · supplementary.pdf]

# Joint Optimization of Tree-based Index and Deep Model for Recommender Systems
## *(Supplementary Material)*

**Han Zhu[1], Daqing Chang[1], Ziru Xu[1,2],\* Pengye Zhang[1]**
[1]Alibaba Group
[2]School of Software, Tsinghua University
Beijing, China
{zhuhan.zh, daqing.cdq, ziru.xzr, pengye.zpy}@alibaba-inc.com

**Xiang Li, Jie He, Han Li, Jian Xu, Kun Gai**
Alibaba Group
Beijing, China
{yushi.lx, jay.hj, lihan.lh, xiyu.xj, jingshi.gk}@alibaba-inc.com

This material has 5 parts in total. Part A gives the detailed proof that $\max_\pi -\mathcal{L}(\theta, \pi)$ given $\theta$ is equivlent to the maximum weighted matching problem of bipartite graph. Part B illustrates the user preference prediction model and the feature constitution used in experiments in detail. Part C contains the detailed settings of experimental study. Part D gives additional experimental results to show the effectiveness of the proposed JTM framework. Part E summarizes related works about tree-based methods for recommendation and extreme classification.

## A Proofs

**Remark 1.** $\max_\pi -\mathcal{L}(\theta, \pi)$ *is essentially an assignment problem to find a maximum weighted matching in a weighted bipartite graph.*

*Proof.* Given the $k$-th item $c_k$ in the corpus $\mathcal{C}$ and the $m$-th leaf node $n_m$ in the tree hierarchy, denote

$$\mathcal{L}_{c_k, n_m} = \sum_{(u,c) \in \mathcal{A}_k} \sum_{j=0}^{l_{max}} \log \hat{p}\left(b_j(n_m)|u; \theta, \pi\right), \qquad (1)$$

where $\mathcal{A}_k = \{(u^{(i)}, c^{(i)})|c^{(i)} = c_k\}_{i=1}^n$ is the training sample set the target item of which is $c_k$.

If we take leaf nodes in $\mathcal{T}$ and items in corpus $\mathcal{C}$ as vertices and the full connection between leaf nodes and items as edges, we can construct a weighted bipartite graph $V$ with $\mathcal{L}_{c_k, n_m}$ as the weight of edge between $c_k$ and $n_m$. Furthermore, we can see that each assignment $\pi(\cdot)$ between items and leaf nodes is equivalent to a matching of $V$. Given an assignment $\pi(\cdot)$, the total loss can be derived as

$$\mathcal{L}(\theta, \pi) = -\sum_{k=1}^{|\mathcal{C}|} \mathcal{L}_{c_k, \pi(c_k)},$$

where $|\mathcal{C}|$ is the corpus size. Therefore, $\max_\pi -\mathcal{L}(\theta, \pi)$ equals to find the maximum weighted matching of $V$. □

# B    Detailed Implementation and Code

The proposed JTM is a joint learning framework composed of two main parts: user preference prediction model and tree learning. In this section, we will give details about both parts.

**User Preference Prediction Model**    Fig 1 shows the user preference prediction model used in offline and online experiments. It takes at most $M$ user-item interactions as input user behavior feature and split them into $N$ time windows in time order. In the experiments, $M$ is set to $69$ and $N$ is $10$. Each of the $10$ time windows contains $[1, 1, 1, 2, 2, 2, 10, 10, 20, 20]$ behaviors respectively (sums up to $69$) from near to far, along the time line. Firstly, we map each item to its corresponding level of target node with tree hierarchy, according to the proposed hierarchical user preference representation. For example, suppose that the initial user behavior sequence is $\boldsymbol{u} = \{c_1, c_2, \cdots, c_m\}$, then we use $\{b_l(\pi(c_1)), b_l(\pi(c_2)), \cdots, b_l(\pi(c_m))\}$ as the user behavior sequence input in level $l$. Then, an embedding layer is applied to get continuous vector representation of the one-hot ID. For embeddings in each time window, we use element-wise average to get one embedding vector, and concatenate them with the target node's embedding as input to the following neural network. The neural network consists of three fully-connected layers, with $128$, $64$ and $24$ hidden units respectively and PReLU [18] as activation function. Finally a binary softmax is used to calculate the probability of user's interest on target node. The corresponding network definition code can be found in `code/user_preference_prediction_model.py` (function `dnn_model_define`). It's worth mentioning that besides the given basic fully connected network, more advanced models like the attention model introduced in TDM[22] are also flexible to be used.

Figure 1: User preference prediction model used in JTM. It can be replaced by arbitrary advanced user preference prediction model and fit into the joint learning framework.

**Tree Learning Algorithm**    In line 5 of the proposed tree learning algorithm (Algorithm 2 in the paper), we use a greedy algorithm with rebalance strategy to solve the sub-problem. Each item $c \in \mathcal{C}_{n_i}$ is firstly assigned to the child of $n_i$ in level $l$ with largest weight $\mathcal{L}_c^{l-d+1,l}(\cdot)$. Then, to guarantee that each child is assigned with no more than $2^{l_{max}-l}$ items, a rebalance process is applied. To promote the stability of tree learning and facilitate the convergence of the whole framework, for nodes that have more than $2^{l_{max}-l}$ items, we keep those items that have the same assignment in level $l$ with the former iteration (i.e., $b_l(\pi'(c)) == b_l(\pi_{old}(c))$) in priority. The other items assigned to the node are sorted in descending order of their weights, and the exceeded part of items are moved to

other nodes that still have redundant space, according to the descending order of each item's weight $\mathcal{L}_c^{l-d+1,l}(\cdot)$. The detailed implementation code are given in `code/tree_learning.py`. Note that the calculation of $\mathcal{L}_c^{s,e}(\pi)$ relies on the aforementioned user preference prediction model and the calculation is omitted in the code.

## C  Experiments

We list some detailed settings of offline experiments in this section, including dataset pre-processing, hyperparameters selection, baseline implementation, hardware and the training schedule.

**Data Pre-processing and Sample Generation**   We conduct offline experiments on two large-scale real-world datasets, Amazon Books and UserBehavior. As introduced in the paper, we only keep users who have no less than 10 records. Following TDM[22], we use item's category information to build initial tree in both datasets. Table 1 summarizes the details of two datasets after pre-processing. From the statistics, we can see that Amazon Books is even sparser than UserBehavior. Amazon Books' corpus size is $35.5\%$ of UserBehavior's, but the number of interactions is less than $9\%$ of UserBehavior's, which brings more challenges to the recommendation problem.

Table 1: Details of the two datasets after preprocessing. Each item is assigned with a unique category. And one record is a user-item pair that represents user's implicit feedback.

|                 | **Amazon Books** | **UserBehavior** |
| --------------- | ---------------- | ---------------- |
| # of users      | 294,739          | 969,529          |
| # of items      | 1,477,922        | 4,162,024        |
| # of categories | 2,637            | 9,439            |
| # of records    | 8,654,619        | 100,020,395      |

We follow the settings of TDM [22] to split the dataset and generate samples.

Considering the user amount, we randomly sample $5,000$ disjoint users to create Amazon Books' validation set and testing set, while $10,000$ disjoint users are selected as UserBehavior's validation and testing set each. Other users in two datasets compose training set accordingly.

The sample generation process for training and validation/testing are different. One sample here means a $(u, c)$ pair mentioned in the paper. For each user in training set, we sliding along the user's behavior sequence to generate samples. In detail, we firstly sort the user-item interactions belonging to the user in ascending order of timestamp Then, a sliding window with size 70 is used to generate samples. Only windows that have at least 6 interactions are kept for training. In each window, the last (with largest timestamp) is the target item, and the others are user behavior features. While for each user in validation and testing set, we take the first half of the user's interactions along the time line as behavior features, and the latter half as ground truth.

Code in `code/data_cutter.py` shows the process of splitting dataset, and the code of sample generation and initial category tree building is provided in `code/data_and_tree_initialize.py`.

**Implementation**   JTM is compared with several baselines, such as Item-CF [14], YouTube product-DNN [4], HSM [10], TDM [22] and DNN. We will give a detailed description of all methods' implementation.

Item-CF is a basic collaborative filtering network. We firstly calculate similarities between items using user behavior sequences. Denote $w_{ij}$ as the similarity between item $c_i$ and item $c_j$. Then we have $p_{uj} = \sum_{i \in N(u) \cap j \in S(i,K)} w_{ji}$ to represent user $u$'s preference over item $c_j$. $N(u)$ is the set of items that have interaction with user $u$, and $S(i, K)$ is a collection of items that most similar to item $c_i$. $K$ is set to $50$ in the experiment.

Unlike Item-CF, other compared methods and JTM all use deep neural network as user preference prediction model. Considering that the training set is quite large, we build Alibaba's open-sourced deep learning platform X-DeepLearning (XDL)[2] in our own GPU cluster to support distributed

model training. YouTube product-DNN uses the inner-product of learnt user and candidate item's vector representations to reflect user preference, while other methods calculate user-item preference with aforementioned neural network in Part B. We implement negative sampling strategy and hierarchical user preference representation in XDL[3]. The code of the user preference prediction model is available in `code/user_preference_prediction_model.py`.

The source code of YouTube product-DNN is provided by TDM[4] and we use it directly.

HSM follows a layer-wise binary classification formulation. Thus its negative samples are the brother nodes of each positive nodes. When prediction, a layer-wise beam-search similar to TDM and JTM is deployed, the only difference is that the score used to choose each level's top-k nodes is the multiplication of each level's conditional probability. We implement it by changing the negative sampling policy and score calculation in prediction in XDL.

TDM has an open-source code implementation of the attention-DNN version in XDL[5]. We change the network define of it to the fully-connected network.

DNN is a variant of TDM without tree index. Like YouTube product-DNN, we directly sample negatives from the item corpus. And in prediction, we linearly scan the whole corpus to find top-k results for each user.

JTM is the proposed joint learning framework of the user preference prediction model and tree index as shown in Algorithm 1 in the paper. And more details and code of these two parts have been introduced in Section B.

**Training Settings**    YouTube product-DNN, TDM, DNN and JTM use negative sampling to train the user preference model. We follow the setting of TDM[22] and use the same negative sampling ratio for all methods. One training sample has $100$ negative samples in Amazon Books and $200$ in UserBehavior.

In the proposed tree learning algorithm (Algorithm 2 in the paper), the layer gap $d$ is a hyperparameter. The choice of $d$ should balance between the approximation and running cost. Suppose that we choose $d = l_{max}$, tree learning algorithm directly solves the maximum matching problem. However, it requires too much calculation to predict preference probability, which is far beyond the capability of GPUs. On the contrary, if we use a too small $d$, for example, $d = 1$, the proposed algorithm would be a bad approximation for the maximum matching problem. In the experiments, we choose a moderate layer gap $d = 7$ for tree learning algorithm and obtain the results shown in the paper. Besides, we have tried other numbers of $d$ around 7, which do not show apparent difference on the convergence and final recommendation accuracy.

All experiments are performed with 20 NVIDIA P100 GPUs. We use Adam [7] to optimize the model and the batch size is set to $30,000$ for each GPU. For Amazon Books, it takes about 35 minutes (about 10 epochs) for a user preference prediction model to converge. UserBehavior has roughly 10 times records of that in Amazon Books and larger negative sampling ratio. Thus it needs about 6 hours (about 4 epochs) to converge. The initial learning rate is set to $0.001$ and it decrease every epoch with a decay rate of $0.9$. Tree learning is conducted once the user preference prediction model training converges. The whole tree learning process takes about 20 and 75 minutes for two datasets respectively.

**Complexity Analysis**    As the user preference model training of different methods is the standard neural network optimization, we focus on the retrieval and tree learning time complexity analysis.

For tree-based methods like HSM, TDM and JTM, layer-wise beam search is used in retrieval. As mentioned in the paper, at most $2k \cdot \log C$ nodes need to be scored with the preference model in a single retrieval process, where $C$ is the size of item set and $k$ is the size of recalled set. Thus the time complexity is $O(2k \cdot \log C)$. YouTube product-DNN and DNN requires to scan all items to find

top-k results, thus the time complexity is $O(C)$. The retrieval time complexities are summarizes in Table 2.

Table 2: Retrieval time complexity comparison.

| Method | Time |
|---|---|
| YouTube product-DNN | $O(C)$ |
| HSM | $O(2k \cdot \log C)$ |
| TDM | $O(2k \cdot \log C)$ |
| DNN | $O(C)$ |
| JTM | $O(2k \cdot \log C)$ |

For the tree learning phase in JTM, Part A proves that the problem equals to a weighted maximum matching problem of bipartite graph, which could be solved by algorithms like Hungarian algorithm. If all the edge weights are known, i.e., $\mathcal{L}_{c,n}$ are known for each item $c$ in the corpus $\mathcal{C}$ and each leaf node $n$ in the tree, Hungarian algorithm can solve the problem with $O(C^3)$ complexity, $C$ is corpus size $|\mathcal{C}|$ (here we suppose the leaf node number in the tree to be close to the corpus size $C$, which could be achieved by carefully control the tree height). However, according to the definition of $\mathcal{L}_{c,n}$, the complexity of calculating all the weights is $O(2CN)$, $N$ is the sample size. Both of the above complexity is not practical for large-scale corpus and large dataset. Thus, we proposed the greedy algorithm to learn the tree. For the proposed greedy algorithm (Algorithm 2 in the paper), if we also suppose that all the used weights are known, the greedy algorithm given in `code/tree_learning.py` can solve the matching problem in $O(C \cdot \log C)$ time complexity. The complexity of calculating all the weights is related to the choice of hyperparameter $d$, as we assign items to non-leaf nodes step-by-step top-down. Given $d$ and the tree height $l_{max}$, the weight calculating complexity is no more than $O(\lceil l_{max}/d \rceil \cdot 2^{d+1} \cdot N)$. Considering that the sample size $N$ is usually larger than the corpus size $C$ in application, the overall complexity of the proposed tree learning is bounded by $O(\lceil l_{max}/d \rceil \cdot 2^{d+1} \cdot N)$.

## D   Additional Results

To explore why the proposed hierarchical user preference representation works, we perform additional experiments on three variants of user preference representation in tree-based model in two datasets.

**Hierarchical User Preference Representation**   Tree-based model samples target nodes from all levels of the tree and uses the concatenation embedding of user behaviors and target node as input. The difference of three variants lies in user behavior features. They all utilize a fixed initial tree described in Section 3.1 of the paper without tree learning. A detailed description of three variants are as follows:

- **TDM** is the basic tree-based model. When dealing with samples from different levels, user behavior feature is totally the same.
- **JTM-HI** is an advanced version of TDM which uses **level-independent feature space**. More specifically, the user behavior features are directly mapped to different embedding spaces when training different levels' models. Compared to TDM, the parameter size increases multiple times according to the height of the tree.
- **JTM-H** is TDM with the proposed **hierarchical user preference representation**. User behaviors in the leaf level are mapped to the nodes in corresponding levels naturally.

From Table 3, we have several observations. JTM-HI outperforms TDM in both datasets, which proves that the level-independent feature space indeed reduces the noise brought by sharing embedding space of user behavior feature in all levels of the tree. JTM-H gets higher performance than JTM-HI with less parameters, which demonstrates that hierarchical user preference representation works well. On the one hand, tree hierarchy provides a natural hierarchical representation. Node embeddings in the same level of tree are homogeneous, thus it's easier to capture latent feature cross in the same level than between leaf and non-leaf levels. On the other hand, with hierarchical user

preference representation, the parameter space of user behavior feature shrinks a lot in upper levels, which partially solves the data sparsity problem.

Table 3: Evaluations of hierarchical representation for user preference model in tree-based models in Amazon Books and UserBehavior ($M = 200$).

| Method | Amazon Books | | | UserBehavior | | |
|--------|-----------|--------|-----------|-----------|--------|-----------|
|        | Precision | Recall | F-Measure | Precision | Recall | F-Measure |
| TDM    | 0.50%     | 7.49%  | 0.88%     | 2.23%     | 10.84% | 3.40%     |
| JTM-HI | 0.53%     | 7.69%  | 0.92%     | 2.40%     | 11.44% | 3.62%     |
| JTM-H  | **0.68%** | **10.45%** | **1.19%** | **2.66%** | **12.93%** | **4.02%** |

In UserBehavior, the recall metric raises from $10.84\%$ to $11.44\%$ with level-independent feature, as a result of feature confusion alleviation between levels. Another recall improvement from $11.44\%$ to $12.93\%$ comes from homogeneity and appropriate granularity features inside each level. The relative improvements are more significant in Amazon Books, as the data sparsity problem is more serious, which can be well solved by the proposed hierarchical user preference representation.

## E Related Work

In real-world applications, the recommendation process usually has two stages: candidate generation and ranking [5, 22, 21]. Model-based large-scale recommendation methods are usually confronted with computational restrictions in the candidate generation stage. To overcome the calculation barrier, there are mainly three kinds of approaches: 1) Pre-calculate item or user similarities and use inverted index to accelerate the retrieval [9]; 2) Convert user preference to distance of embedding vectors, and use approximate kNN search in retrieval [4]; 3) Use tree or ensemble of trees to perform efficient retrieval [22].

Industrial recommender systems typically adopt vector kNN search to achieve fast retrieval, e.g., YouTube video recommendation [4, 2], Yahoo news recommendation [11] and extensions that use recurrent neural network to model user behavior sequence [6, 15, 17]. Such approaches use either traditional deep neural network (DNN) or recurrent neural network (RNN) to learn user and item's embedding representations based on various user behavioral and contextual data. However, due to the dependence of approximate kNN search index structures in retrieval, user preference models that use attention network or cross features [20, 19, 3] are challenging to be applied.

Tree-based methods are also studied and adopted in real-world applications. Label Partitioning for Sublinear Ranking (LPSR) [16] uses k-means clustering with data points' features to learn the tree hierarchy and then assign labels to leaf nodes. In the prediction stage, the test sample is passed down along the tree to a leaf node according to its distance to each node's cluster center, and the 1-vs-All base classifier is used to rank all labels belonged to the retrieved leaf node. Partitioned Label Trees (Parabel) [13] also use recursive clustering to build tree hierarchy, but the tree is built to partition the labels according to label similarities. Multi-label Random Forest (MLRF) [1] and FastXML [12] learn an ensemble of sample partitioning trees (a forest), and a ranked list of the most frequent labels in all the leaf nodes retrieved from the forest is returned in prediction. MLRF optimizes the Gini index when splitting nodes, and FastXML optimizes a combined loss function including a binary classification loss and a label ranking loss. In all the above methods, the tree structure keeps unchanged in training and prediction once built, which is hard to completely adapt the retrieval model dynamically.

The previous work TDM [22] introduces a tree-based model for large-scale recommendation differentiated from existing tree-based methods with a max-heap like user-node preference formulation. In TDM, tree is used as a hierarchical index [8], and an attention model [20] is trained to predict user-node preference. Different from most tree-based methods where non-leaf nodes are used to route decision-making to leaves, TDM explicitly formulates user-node preference for all the nodes to facilitate hierarchical beam search in the tree index. Despite achieving remarkable progress, the joint optimization problem of index and model is not well solved yet as that the proposed alternatively learning method of model and tree has different objectives.

## Footnotes

\*The work is done when she was a student intern in Alibaba Group

[2] `http://github.com/alibaba/x-deeplearning`

[3]Note that hierarchical user preference representation is only used in JTM.

[4]`http://github.com/alibaba/x-deeplearning/tree/master/xdl-algorithm-solution/TDM/script/tdm_ub_vector_ubuntu`

[5]`http://github.com/alibaba/x-deeplearning/tree/master/xdl-algorithm-solution/TDM/script/tdm_ub_att_ubuntu`