[Reviews · NeurIPS 2019]

Reviewer 1



* This study is solving an important problem in the area of recommendation systems, namely, candidate set generation * The idea is benchmarked against popular metrics (precision, recall, f-measure) over two public datasets, with offline and online experiments. * The baselines benchmarked against are also popular and standard approaches (DNN, ITEM-CF). The results presented for this work beats the benchmarks by a good deal, and in particular, the online test results are very good. * I am not sure if the novelty of this paper is suitable for NeurIPS. It is an incremental improvement to an existing model (TDM) by doing an additional optimization step. The resulting improvement is impressive, though, and it feels like this would be more applicable to an applied data science conference such as KDD or WWW. * There are portions of the paper that are difficult to follow, especially in the explanation of the Joint Optimization Framework. The explanation of TDM in Section 2.1 is helpful, but it would be even more helpful to have a direct comparison between the tree building steps between TDM and the new proposed method. For example, having a side-by-side comparison of Algorithms1 & 2 with its TDM predecessor would go a long way in understanding detailed differences. * Moving to iterative joint learning would likely introduce challenges for training time (or potentially challenging of infrastructure for supporting optimizing tree hierarchy). It would be good if the authors could provide some tradeoff or comparison there. Maybe that could explain why the paper did not mention whether it has been deployed to production (although online A/B results were shown). * It would be good to talk about some of the more practical aspects, such as how many levels of the tree is chosen and how sensitive is the algorithm to these kinds of parameters? * Figure 2: it seems that the Clustering algorithm outperforms JTM in the first few iterations, so would be curious about the intuitive explanation why that’s the case. * Although the paper mentioned that an A/B experiment were performed to evaluate on CTR & RPM (no statistical significance reported), but no where in the paper mentioned whether the method were finally deployed to the full production system. It would be good to have clarity on this in the paper.

Reviewer 2



The paper proposes a joint model to simultaneously learn item tree index and user representation that support efficient retrieval. The tree learning algorithm is based on the maximum weight matching in the bipartite graph for which an approximate algorithm is proposed. Authors further propose to iteratively leverage the tree index to infer a hierarchical user representation that is shown to improve recommendation accuracy. The paper is well written and addresses an important problem in collaborative filtering where fast retrieval becomes critically important at scale. I think the proposed approach is sound and I particularly like the hierarchical tree-based user presentation. It is quite different than most of the proposed approaches that are either neighbour-based or use inner product latent representations. The experimental section provides a thorough comparison using large public datasets. It is also useful to see that the method performs well in a live production setting. Here, I would have also liked to see training and inference times given that even with the proposed approximations Algorithm 2 can be expensive. I think the authors need to quantify this overhead and compare it to other approaches, particularly since the method is targeted towards very large datasets.

Reviewer 3



Originality: Authors propose to extend the tree-based deep model (TDM) for recommendations by jointly optimizing the tree structure. The idea of joint learning itself is somewhat straightforward. The proposed algorithm is also somewhat straightforward, but it has the advantage of being simple and clear, which will allow future research to easily build upon it. Quality: The contribution of this paper is mostly empirical, and the performance of the proposed approach is impressive in both online and offline evaluations. Authors do a great job in making sure relevant baselines are included in the offline analysis, which allows us to measure the impact of each design choice (for ex: joint learning vs. only learning deep NN parameters). Offline experiments are conducted on public datasets, which shall be replicated in the future. Clarity: The paper is clearly written. Authors provide sufficient literature survey and the paper is mostly self-contained. However, it could've been nicer if the network structure of the proposed model was described. Significance: The proposed method is a straightforward but clearly useful extension of TDM. It provides significant improvements over TDM or other strong baseline models, and therefore has a good chance of being widely used in large-scale recommender systems.

[Author Response · NeurIPS 2019]

| | UserBehavior |
|---|---|
| # of users | 969,529 |
| # of items | 4,162,024 |
| # of categories | 9,439 |
| # of interactions | 100,020,395 |
| # of samples | **tens of billions** |

Figure 1: Dataset size summary

Figure 2: Time cost

Figure 3: Parameter sensitivity

Thank all the reviewers for your affirmation and insightful comments. Here is the response to some of your concerns.

**To Reviewer #1 & #2**

As that the time cost is a common concern, we do further experiments to quantify the time cost (detailed time complex-
ity analysis is given in Line 121 of the supplementary material). Here we evaluate the running time with UserBehavior
dataset, the size of which is given in Figure 1. The time cost of one iteration (including one epoch model training and
once tree learning) time w.r.t. the GPU card number are given in Figure 2. We can observe that the JTM's running time
decreases rapidly with the increase of GPU usage, which indicates that JTM is scalable enough for industrial applica-
tions, though involves iterative training. Especially, the time cost of JTM approaches to TDM with the increase of card
number, while achieving significantly better performance. **JTM has already been fully deployed in the production**
**system. The model and tree structure are daily updated, serving billions of impressions every day.**

**To Reviewer #1**

**Q**: It would be helpful to have an intuitive explanation between the proposed JTM and the existing TDM. It would be
helpful to have a direct comparison between the tree building steps between JTM and TDM.

JTM addresses the key problem in large-scale recommendation, i.e., how to optimize user representation, user prefer-
ence prediction and tree structure under a global objective. JTM proposes a **unified framework** to integrate the opti-
mization of these three key factors, while they are optimized separately in TDM. The contribution and improvements
are affirmed by the experimental results. JTM optimizes the tree structure by solving the combinatorial optimization
problem $\max_\pi -\mathcal{L}(\theta, \pi)$ under the unified framework, while TDM uses an intuitive clustering to learn the tree.

**Q**: It would be good to talk about some of the more practical aspects and the algorithm's parameter sensitivity.

Some of the detailed practical aspects are given in the supplementary material considering the space limit. For example,
we use complete binary tree in the experiments, the height of which only depends on the size of item corpus. As for the
algorithm's parameter sensitivity, we do additional experiments to evaluate the sensitivity w.r.t. different tree learning
gap $d$ (used in Algorithm 2) and the results are in Figure 3. The results indicate that JTM is fairly stable w.r.t. $d$.

**Q**: It seems that the clustering algorithm outperforms JTM in the first few iterations, so would be curious about the
intuitive explanation why thats the case.

In JTM, the proposed approximate tree learning algorithm involves a *lazy strategy*, i.e., try to reduce the degree of
tree structure change in each iteration (details are in Line 39-40 of the supplementary material). That's why the recall
metric of JTM doesn't increase as quickly as clustering in the first two iterations. However, the recall metric of JTM
keeps increasing and converges to much better final results compared to clustering.

**To Reviewer #3**

**Q**: It would've been useful to compare against variants of TDM other than the basic DNN version. It would be
interesting to understand how the gap parameter $d$ affects the performance.

We use the basic DNN version of TDM (in Figure 1 of the supplementary material) in all offline comparison, since
JTM's merits do not rely on specific network structure. To verify this, we do further experiments with TDM's attention-
DNN variant. The recall results of TDM and JTM in UserBehavior dataset (introduced in Figure 1) are **13.07%** and
**18.85%** respectively, which is an even more significant improvement. As for the gap parameter $d$, we evaluate several
values other than 7. The results in Figure 3 indicates that the performance is not sensitive w.r.t. the choice of $d$.

[Meta-Review · NeurIPS 2019]

The review scores were somewhat borderline, but overall slightly above the acceptance threshold. There was some disagreement among the reviewers, following which a discussion was initiated. The rebuttal largely addresses the concerns of R1 (the most negative review), and in the metareviewer's opinion does a reasonable job of addressing these concerns, which are mostly clarifications regarding the performance of the algorithm. Positively, the reviewers mostly concur that the method, while fairly straightforward, offers significant improvements over existing techniques. After discussion there was some positive movement in review scores resulting in a positive consensus among reviewers.